# Variations in Prehospital Analgesic Use Based on Pain Etiology

**DOI:** 10.3390/biomedicines13071620

**Published:** 2025-07-01

**Authors:** Nikolina Marić, Radojka Jokšić-Mazinjanin, Aleksandar Đuričin, Luka Ivanišević, Goran Rakić, Zoran Gojković, Mirka Lukić Šarkanović, Milena Jokšić Zelić, Lucija Vasović, Velibor Vasović

**Affiliations:** 1Institute for Emergency Medical Services Novi Sad, 21000 Novi Sad, Serbia; maric1992@gmail.com (N.M.); aleksandar.djuricin@mf.uns.ac.rs (A.Đ.); 2Medical Faculty, University of Novi Sad, 21000 Novi Sad, Serbia; 015265@mf.uns.ac.rs (L.I.); goran.rakic@mf.uns.ac.rs (G.R.); zoran.gojkovic@mf.uns.ac.rs (Z.G.); mirka.lukic-sarkanovic@mf.uns.ac.rs (M.L.Š.); velibor.vasovic@mf.uns.ac.rs (V.V.); 3Pediatric Surgery Clinic, Institute for Child and Youth Health Care Vojvodina, 21000 Novi Sad, Serbia; 4University Clinical Center of Vojvodina, 21137 Novi Sad, Serbia; 5Health Center Bečej, 21220 Bečej, Serbia; milenajoksiczelic@gmail.com; 6Institute for Pulmonary Diseases of Vojvodina, 21204 Sremska Kamenica, Serbia; vasovicl720@gmail.com; 7Academy of Medical Sciences, Serbian Medical Society, 11000 Belgrade, Serbia

**Keywords:** pain, bone fractures, myocardial infarction, angina pectoris, chest pain, analgesia

## Abstract

**Background/Objectives**: Pain is the most frequently reported symptom in over 90% of patients presenting with traumatic injuries, and three-quarters of patients are discharged from emergency departments experiencing moderate to severe pain. The objective of this study was to compare the frequency of analgesic administration between patients with chest pain presumed to be of cardiac origin and those with suspected bone fractures as well as to assess whether significant differences exist between these two groups. **Methods**: A retrospective, observational study was conducted. Patients were categorized into two groups: Group 1—patients with angina pectoris, acute myocardial infarction, or non-specific chest pain; and Group 2—patients with a preliminary diagnosis of bone fracture made by the attending physician at the scene. **Results**: A total of 1189 patients were included in this study, with 503 (42.3%) in Group 1 and 686 (57.7%) in Group 2 (χ^2^ = 28.166; *p* < 0.001). Analgesic administration was significantly more frequent among patients in Group 1 than in Group 2 (χ^2^ = 23.187; *p* < 0.001). Within Group 1, the highest rate of analgesic use was recorded in patients diagnosed with acute myocardial infarction. In Group 2, analgesics were administered to 36.4% of patients with suspected trunk bone fractures, while only 7.1% of patients with suspected cranial fractures received analgesic therapy. Pain intensity scores were not available for either group. **Conclusions**: The administration of analgesic treatment was significantly more common among patients presenting with chest pain of presumed cardiac origin than among those with suspected bone fractures, including fractures involving multiple body regions.

## 1. Introduction

Emergency departments (EDs) across Europe manage approximately 38 million cases annually involving injuries of diverse etiologies, with an estimated 5 million patients requiring hospitalization [1]. Pain is the predominant symptom in over 90% of these patients. Despite its high prevalence, studies have indicated that up to 75% of patients are discharged from EDs with ongoing moderate to severe pain [2,3].

Evidence suggests that nearly two-thirds of trauma patients wait an hour or longer before receiving analgesia and that the choice of analgesic agent(s) frequently does not align with the severity of the reported pain [4]. Commonly administered analgesics in both prehospital and emergency settings in Europe include paracetamol, non-steroidal anti-inflammatory drugs (NSAIDs), nitrous oxide, and opioids [5,6]. Among these, opioids are the most frequently used for managing acute trauma-related pain; however, their effectiveness has been increasingly questioned [7,8]. Animal studies suggest that opioids may not only be ineffective but could also exacerbate pain and prolong its duration [9].

Inadequate and delayed management of acute pain in trauma patients can result in substantial physiological and psychological consequences [10]. Poorly managed acute pain has been associated with impaired early mobilization, suboptimal rehabilitation outcomes, and an elevated risk of transition to chronic pain syndromes [11,12].

The situation is comparable in cases of chest pain. Approximately 1% of patients who consult a general practitioner or contact emergency medical services do so because of chest pain [13]. Among these patients, 2–4% have pain of cardiac origin, requiring further diagnostic evaluation or treatment [14]. Current clinical guidelines for the management of chest pain of cardiac origin emphasize the use of antiplatelet and anticoagulant therapies. However, these guidelines offer limited direction on pain management despite the fact that unrelieved pain may provoke sympathetic stimulation and vasoconstriction, thereby increasing myocardial oxygen demand and exacerbating cardiac workload [15].

For many years, opioids—particularly morphine—have been regarded as the first-line analgesic in the management of chest pain in patients with suspected acute coronary syndrome (ACS) [15,16]. Nevertheless, due to adverse effects such as nausea, vomiting, hypotension, and bradycardia [17]—as well as concerns regarding potential delays in the efficacy of antiplatelet agents—clinicians in prehospital and emergency department settings have contributed to a decline in the routine use of morphine as the analgesic of choice [15,16].

Effective pain control remains a key priority in prehospital care. According to the most recent guidelines, timely and effective pain control must be ensured for all patients experiencing pain, ideally at the first point of contact with emergency medical services (EMS). The primary objectives should be to reduce pain intensity, maintain patient functionality, and minimize adverse effects. The use of standardized protocols at the prehospital level is strongly recommended. Emergency care protocols should, among other elements, include an initial assessment of the presence and intensity of pain using validated pain scales, the use of both pharmacological and non-pharmacological pain management strategies, ongoing monitoring and re-evaluation of pain following analgesic administration, and ensuring the accurate transfer of pain-related information during handover to hospital-based teams [18]. Analgesic therapy should be administered promptly to patients experiencing pain, and the selection of analgesics should be tailored to the intensity and etiology of the pain, with careful consideration of potential side effects. Physicians working in emergency medicine must have access to a broad range of pharmacologic options and must be well-versed in the clinical indications, contraindications, methods of administration, and potential adverse effects of analgesic agents [19,20].

The aim of this study was to evaluate the frequency of analgesic administration by emergency medical service (EMS) teams at the prehospital level in patients presenting with chest pain of presumed cardiac origin, compared to those with suspected bone fractures. Additionally, this study seeks to determine whether there is a statistically significant difference in the frequency of prehospital pain management between these two patient groups.

## 2. Materials and Methods

### 2.1. Study Design

A retrospective, observational study was conducted over a six-month period, from 1 January to 30 June 2024. Data were collected at the prehospital level from the Institute of Emergency Medical Services Novi Sad (IEMS NS).

### 2.2. Data Source

IEMS NS operates within a region inhabited by 368,967 people, according to the 2022 national census. The institute comprises a central dispatch center responsible for receiving and triaging emergency calls and coordinating the deployment of nine EMS teams. Each team is staffed by an emergency medicine specialist or general physician, a medical technician, and a driver trained in basic life support.

All EMS teams are equipped to perform advanced life support, manage trauma, and diagnose ACS. Despite the presence of physicians in EMS teams, the range of analgesic medications available for prehospital use remains limited. The list of medications authorized for EMS use is determined by the national health insurance fund. Currently, EMS teams only have access to a small selection of injectable analgesics, including two opioids (tramadol and morphine), two non-steroidal anti-inflammatory drugs (NSAIDs: diclofenac and ketoprofen), and metamizole sodium. Other injectable analgesic agents are not available for prehospital administration.

Patient data for the study were obtained from field protocols documented by all nine EMS teams. The following variables were recorded: patient age, sex, vital signs (systolic blood pressure [SBP], peripheral oxygen saturation [SpO_2_], and heart rate), prehospital working diagnosis, the administration of analgesic therapy, and the type of analgesic used, if applicable. All data were stored in electronic or paper format and treated as strictly confidential.

### 2.3. Study Population

This retrospective study included all patients aged 18 years and older treated by EMS teams during the defined period for chest pain or suspected bone fractures, who were subsequently transported to either the Institute for Cardiovascular Diseases of Vojvodina (ICVDV) or the Emergency Department of the University Clinical Center of Vojvodina (UCCV).

Patients were divided into two groups based on their prehospital working diagnoses, coded according to the 10th revision of the International Classification of Diseases (ICD-10).

Group 1: Patients with a working diagnosis of chest pain (R07), angina pectoris (I20), or suspected acute myocardial infarction (I21) based on electrocardiographic changes (either ST-segment elevation or depression).Group 2: Patients who had sustained trauma and were assigned a working diagnosis of bone fracture by the EMS physician following the initial assessment. These diagnoses included S02 (fractures of the skull and facial bones), S12 (cervical spine fractures), S22 (rib, sternum, and thoracic spine fractures), S32 (lumbar spine and pelvic fractures), S42 (shoulder and upper arm fractures), S52 (forearm fractures), S62 (wrist and hand fractures), S72 (femoral fractures), S82 (lower leg and ankle region fractures), and S92 (foot fractures excluding the ankle), as per ICD-10. All patients in Group 2 were transported to the Emergency Center of the University Clinical Center of Vojvodina (UKCV).

Each group was further stratified into subgroups based on the assigned working diagnoses, allowing for a more detailed analysis of analgesic administration relative to the underlying clinical condition.

Group 1 was divided into three diagnostic subgroups:Chest pain (R07);Angina pectoris (I20);Acute myocardial infarction (I21).

Group 2 was divided into four diagnostic subgroups:
Cranial fractures, including patients with diagnoses S02 (skull and facial fractures) and S12 (cervical spine fractures);Trunk fractures, including patients with diagnoses S22 (rib, sternum, and thoracic spine fractures) and S32 (lumbar spine and pelvic fractures);Extremity fractures, including diagnoses S42, S52, S62, S72, S82, and S92 (fractures of the upper and lower limbs);Fractures involving two or more anatomical regions are defined as patients assigned two or more of the S-codes from above.

Patients with chest pain who were not transported to ICVDV—i.e., those deemed to have non-cardiac pain and treated at home—were excluded from the study. Patients presenting with other types of injuries, such as contusions, open wounds, dislocations, or sprains, were also excluded.

### 2.4. Data Processing

A total of 1189 patients were included in the study and categorized into two groups based on the predefined criteria. Group 1 comprised 503 patients (42.3%), while Group 2 included 686 patients (57.7%).

The analysis involved comparing group sizes and examining demographic variables (sex and age) and vital parameters between the two groups. The frequency of analgesic administration was assessed in both groups. In addition, we examined whether analgesic use was associated with sex, age, vital signs, and specific diagnostic codes or anatomical regions.

The types of analgesics administered were also analyzed to identify potential differences in drug selection between the groups. Analgesics were categorized as opioids, NSAIDs, metamizole sodium, or paracetamol. Special attention was given to pain assessment scales; however, we also recorded whether such scales were applied and whether their values were documented in the field protocols.

### 2.5. Statistical Analysis

All statistical analyses were performed using IBM SPSS Statistics, version 23. Numerical variables are expressed as the mean ± standard deviation or as the median with interquartile range (P25–P75), depending on the distribution of the data. Categorical variables are presented as absolute frequencies with corresponding percentages (N, %) and were analyzed using the chi-square test or Fisher’s exact test when subgroup sizes were sufficiently small. Differences between two groups of continuous variables were assessed using either the independent samples *t*-test or the Mann–Whitney U test, depending on the normality of the distribution, which was evaluated using the Kolmogorov–Smirnov test.

A *p*-value of less than 0.05 was considered statistically significant for all tests. The results are presented in tabular form.

## 3. Results

A total of 1189 patients were included in the study. The distribution of patients between the two groups showed a statistically significant difference, with a higher proportion in Group 2 (χ^2^ = 28.166; df = 1; *p* < 0.001).

No significant gender difference was observed in the overall sample, although the proportion of male patients (52.6%) was slightly higher than that of female patients (47.4%). However, a statistically significant gender difference was noted within the groups: Group 1 had a higher proportion of male patients (χ^2^ = 32.066; df = 1; *p* < 0.001), whereas Group 2 had a higher proportion of female patients (χ^2^ = 5.971; df = 1; *p* = 0.014).

Patients in Group 1 were significantly older than those in Group 2 (t_1171_ = −4.741; *p* < 0.001) and had significantly higher systolic blood pressure values (Z = −2.483; *p* = 0.013). No statistically significant differences were observed between the groups in terms of SpO_2_ or heart rate (*p* > 0.05) (Table 1).

The data presented in Table 2 reveal that the overall frequency of analgesic administration across the entire sample was 35.8%. This proportion was significantly lower than the 64.2% of patients who did not receive analgesia (χ^2^ = 95.515; df = 1; *p* < 0.001). A statistically significant difference was observed between the two groups, with a notably higher proportion of patients in Group 1 receiving analgesic therapy compared to Group 2 (60.2% vs. 17.9%; χ^2^ = 23.187; df = 1; *p* < 0.001).

Within-group analysis further indicated a significantly higher proportion of patients receiving analgesia in Group 1 than those who did not (χ^2^ = 21.091; df = 1; *p* < 0.001). In contrast, in Group 2, significantly fewer patients received analgesic therapy than those who did not receive such therapy (χ^2^ = 282.216; df = 1; *p* < 0.001).

The chi-square test of independence confirmed a statistically significant association with the likelihood of receiving analgesic therapy among the two groups (χ^2^ = 224.216; df = 1; *p* < 0.001).

In Group 1, the predominant working diagnosis was angina pectoris (ICD-10 code I20; N = 220), whereas the least frequent diagnosis was unspecified chest pain (R07; N = 68). Among patients diagnosed with acute myocardial infarction (I21), nearly 75% received analgesic therapy, a proportion that was statistically significant (*p* < 0.001). More than half of the patients diagnosed with angina pectoris were administered analgesic. In contrast, less than half of those with a diagnosis of unspecified chest pain were treated with analgesics. Nevertheless, the frequency of analgesic administration across all diagnostic categories within Group 1 was significantly higher than in Group 2.

In Group 2, the most frequently assigned working diagnosis was suspected fracture of the extremities, followed by suspected fractures involving multiple anatomical regions. Suspected trunk fractures were the least frequent diagnosis. Analgesic therapy was most frequently administered to patients with suspected trunk fractures and least frequently to those with suspected cranial fractures (*p* < 0.001). Among patients with suspected fractures involving two or more anatomical regions, only approximately one in nine received analgesic treatment (Table 3).

Further within-group analysis in Group 1 revealed that the proportion of male patients receiving analgesia (63.2%) was significantly higher than those who did not (36.8%) (χ^2^ = 29.585; df = 1; *p* < 0.001). Regarding female patients in Group 1, although a higher proportion received analgesia (55.3%) compared to those who did not (44.7%), this difference was not statistically significant (*p* > 0.05). In the subgroup of patients diagnosed with angina pectoris, a significantly higher proportion of male patients received analgesic therapy compared to those who did not (χ^2^ = 7.119; df = 1; *p* = 0.008). Additionally, patients who received analgesia exhibited significantly elevated systolic blood pressure values relative to those who did not (Z = −2.311; *p* = 0.021). Among patients with acute myocardial infarction, analgesic administration was significantly more frequent in both male (χ^2^ = 25.352; df = 1; *p* < 0.001) and female (χ^2^ = 16.781; df = 1; *p* < 0.001) patients. In contrast, among individuals with non-specific chest pain (R07), no statistically significant differences in any demographic or clinical parameters were observed between those who received analgesia and those who did not (*p* > 0.05) (Table 4).

When analyzing Group 1 as a whole, no statistically significant association between sex and analgesic administration was found (χ^2^ = 2.714; df = 1; *p* = 0.099). Similarly, no sex-based differences were observed within the acute myocardial infarction subgroup (χ^2^ = 0.078; df = 1; *p* = 0.779) or the non-specific chest pain subgroup (χ^2^ = 0.005; df = 1; *p* = 0.942). However, a significant sex disparity was noted in the angina pectoris subgroup, in which male patients were more likely to receive analgesic therapy than female patients (χ^2^ = 6.075; df = 1; *p* = 0.014).

In Group 2, more than 80% of male and female patients did not receive analgesia. These proportions were statistically significantly higher than those who received analgesic therapy (male: χ^2^ = 140.453; df = 1; *p* < 0.001; female: χ^2^ = 142.296; df = 1; *p* < 0.001).

Additional analyses explored the physiological characteristics associated with analgesic administration. Within Group 1, patients who received analgesic therapy exhibited significantly higher systolic blood pressure values (Z = −2.314; *p* = 0.021). In Group 2, patients who received analgesia were significantly older (t = −2.234; *p* = 0.026) and demonstrated significantly higher heart rate values (Z = −1.970; *p* = 0.049) when compared to their counterparts who did not receive analgesia (Table 5). In Group 2, across both sexes, patients with extremity and multiple-region fractures were significantly less likely to receive analgesia. Among males, this trend was observed in patients with extremity fractures (χ^2^ = 43.758; df = 1; *p* < 0.001) and those with fractures involving multiple anatomical regions (χ^2^ = 92.627; df = 1; *p* < 0.001). The same pattern held for female patients (χ^2^ = 58.260; df = 1; *p* < 0.001 and χ^2^ = 88.971; df = 1; *p* < 0.001, respectively). Notably, patients with multiple-region fractures who received analgesia had significantly higher heart rate values than those who did not (Z = −2.079; *p* = 0.038) (Table 5).

No statistically significant association between sex and analgesic administration was found in Group 2 overall (χ^2^ = 0.726; df = 1; *p* = 0.394) nor within the subgroups of extremity fractures (χ^2^ = 0.380; df = 1; *p* = 0.394) or multiple-region fractures (χ^2^ = 0.439; df = 1; *p* = 0.537). Due to the small subgroup sizes, Fisher’s exact test was used for trunk fractures, which also did not reveal any sex-based differences in analgesic use (*p* = 0.652). For cranial fractures, statistical comparison by sex was not feasible, as none of the male patients in this subgroup received analgesia.

Among all patients diagnosed with angina pectoris (I20), tramadol was the sole analgesic administered. The majority of these patients received a 50 mg dose (57.5%), while the remainder were treated with a 100 mg dose; in one case, a cumulative dose of 150 mg was recorded. Among patients with a working diagnosis of acute myocardial infarction (I21), 100 mg of tramadol was more frequently administered than the 50 mg dose. Additionally, one patient in this subgroup received morphine. Tramadol was also the predominant analgesic used among patients with a diagnosis of unspecified chest pain (R07), with the 50 mg dose being the most common—a pattern consistent with that observed in cases of angina pectoris. Notably, only one patient in Group 1 received an NSAID (Table 6).

For patients presenting with suspected fractures in Group 2, NSAIDs were administered more frequently than tramadol across all anatomical regions. The use of metamizole sodium was also prevalent, especially among patients with extremity fractures. Tramadol at a dose of 100 mg was more frequently prescribed in cases involving fractures of two or more body regions (Table 7).

Overall, tramadol was the most commonly used analgesic in both study groups. In Group 1, 50.8% (154/303) of patients receiving analgesia were given 50 mg of tramadol, while 48.2% (146/303) received 100 mg of tramadol. In contrast, among Group 2 patients, 20.5% (25/122) received 100 mg of tramadol, and 19.7% (24/122) received 50 mg of tramadol. Metamizole sodium was used in 18.0% of Group 2 cases and was combined with 50 mg of tramadol in one patient in Group 1. NSAIDs were administered in only 1 case in Group 1 but were administered in 41.8% (51/122) of Group 2 patients.

Pain intensity scores were not available for any patient at the prehospital level.

## 4. Discussion

This study examined the frequency of analgesic administration in two distinct patient groups to determine whether the approach to acute pain management differs depending on pain etiology. The findings demonstrate that patients with pain of presumed cardiac origin were significantly more likely to receive analgesia compared to those with suspected bone fractures. Importantly, no pain assessment scales were used by EMS teams in the field.

Pain remains the leading cause of patient presentation to urgent centers (UCs) in the United States [21] as well as the most frequent reason for contacting emergency medical services [22]. Approximately 10.2% of emergency calls are attributed to chest pain [23], while injuries account for around 21.8% of cases [24]. Over the past decade, research on analgesia within UC settings has increased substantially, frequently challenging longstanding paradigms and prompting reconsideration of current pain management strategies [25], both in UC and prehospital environments. The Montreal Declaration of the International Pain Summit identifies pain management as a global health priority and highlights the ethical obligation of all health care providers—including those operating in prehospital environments—to ensure effective pain relief [26]. Despite growing evidence and the expansion of clinical guidelines, oligoanalgesia (defined as the undertreatment of pain) remains a persistent and significant concern in clinical practice [27].

The proportion of patients receiving analgesic therapy in the present study was notably higher among those presenting with chest pain compared to individuals with injuries and suspected fractures. Lord et al. reported that approximately 45% of patients experiencing pain at the prehospital setting did not receive analgesic treatment [28], a finding consistent with our observations regarding chest pain management but diverging significantly from our findings related to injuries and suspected fractures. Previous studies have indicated that approximately 70% of patients with injuries receive analgesia in the prehospital setting [29]. Similarly, Magnusson et al. (2021) examined analgesic administration in patients presenting with knee injuries, chest pain, and abdominal pain, demonstrating that 34.8% of patients with chest pain and 58.3% of those with knee injuries received prehospital analgesia [30]. In New Zealand, clinical performance targets recommend that 90% of patients with chest pain of presumed cardiac origin should receive analgesia within the first hour of contact with emergency services—a goal that is generally achieved in practice [31]. In contrast, the present study revealed considerably lower rates of analgesia administration, particularly among patients with suspected fractures, of whom only 17.9% received analgesic therapy, compared to 60.2% of patients with chest pain.

Analgesic therapy was least frequently administered to patients with suspected fractures of the head and face, as well as those with suspected fractures involving multiple anatomical regions—populations in which more intensive pain management would be clinically warranted due to the high probability of polytrauma [32]. This discrepancy may be attributed to the clinical complexity of these patients, including the high prevalence of hemodynamic instability and the prioritization of life-saving interventions, resulting in the under-recognition or deprioritization of pain management in the prehospital context. In contrast, among patients presenting with chest pain, those diagnosed with acute myocardial infarction (I21) based on electrocardiographic findings received analgesic therapy most frequently (72.1%). In contrast, those with nonspecific chest pain without electrocardiographic changes (R07) had the lowest analgesia rates (41.2%). This difference may be explained by earlier treatment protocols for acute myocardial infarction, which recommend the routine administration of morphine, oxygen, nitroglycerin, and aspirin (MONA) [33]. Although current guidelines for the management of acute coronary syndrome place greater emphasis on anticoagulation and antiplatelet therapy over routine analgesia [15], the influence of previous recommendations likely continues to shape clinical practice at the prehospital level.

Bone fractures represent a major global public health concern, with a higher incidence reported among males; moreover, their occurrence continues to rise [34]. In contrast to these findings, the present study identified a statistically significantly greater proportion of women with suspected fractures. Although male patients were statistically over-represented in Group 1, which is consistent with existing epidemiological literature, analgesic therapy was not consistently administered across all subgroups. While males predominated in all three diagnostic categories (angina pectoris, myocardial infarction, and chest pain), only those with angina pectoris were significantly more likely to receive analgesia. These findings align with previous studies reporting a higher incidence of ACS in men, who tend to present at a younger age compared to women [35]. Neumann et al., in a comprehensive analysis of ACS cases in Germany over a ten-year period, reported that women accounted for approximately 36% of patients, a proportion consistent with the findings of the present study [36]. Previous research has consistently indicated that women are less likely to receive analgesia, are prescribed less potent analgesics, and experience longer delays in receiving pain management compared to men [37].

Similar patterns have been observed among trauma patients. Wimbish et al. demonstrated that opioid analgesics were administered more frequently to male trauma patients than to females [38]. In the present study, men in Group 1 (chest pain) were significantly more likely to receive analgesic therapy compared to women. Conversely, among patients with suspected fractures (Group 2), analgesia was administered more frequently to women. Subgroup-level analyses revealed additional complexity. While women with chest pain or acute myocardial infarction were, in some cases, more likely to receive analgesia than men, a reverse trend was observed in trauma-related cases. Specifically, in Group 2, women were more frequently administered analgesia for cranial and extremity fractures than their male counterparts. These observations suggest that both sex and pain etiology may influence prehospital analgesic decision making, underscoring the need for more standardized, evidence-based protocols that mitigate implicit biases and ensure equitable care delivery across patient populations.

Men are generally less sensitive to pain than women, a phenomenon attributed to genetic factors, body composition, sex hormones, and various psychosocial influences. Cardiac-related chest pain most commonly results from ischemic episodes that activate both chemoreceptors and mechanoreceptors within the heart. Myocardial ischemia triggers the release of numerous biochemical mediators, which stimulate sympathetic and vagal afferent nerve fibers.

Ischemic chest pain may present as either typical or atypical. Typical pain is characterized by discomfort in the chest radiating to the left arm and is mediated by sympathetic afferent fibers. These fibers project to the spinal cord via thoracic and lower cervical segments, synapsing in the dorsal horn and subsequently relaying signals through the thalamus to the cerebral cortex. Atypical pain, more frequently reported in women, manifests as discomfort in the back, neck, or lower jaw. This presentation is mediated primarily by vagal afferent fibers, which transmit impulses through the nucleus of the solitary tract and the C1–C2 spinal segments.

An individual’s psychological state can also influence the perception of cardiac-related pain via neural pathways involving the amygdala [39]. Conversely, tissue injury due to trauma leads to the local release of inflammatory mediators, which activate nociceptors. These nociceptive impulses are conveyed via afferent fibers to the dorsal roots of the spinal cord, from where they ascend to the thalamus and subsequently to the primary and secondary somatosensory cortices [40].

Sex-based differences in pain perception are well-documented. Women exhibit greater somatic awareness compared to men, resulting in heightened pain sensitivity but lower specificity for symptoms such as chest pain. Women are also more likely to present with atypical pain localization [41].

In our study, we found that women with a working diagnosis of acute myocardial infarction (I21) were more frequently administered analgesic therapy than men. This finding likely reflects both heightened pain sensitivity and the presence of electrocardiographic changes. In contrast, among patients diagnosed with angina pectoris (I20), analgesia was more frequently administered to men. This may be attributed to the typical presentation of chest pain observed in this subgroup. These findings underscore the importance of considering gender differences in pain perception and treatment in clinical practice.

Effective pain management remains a fundamental aspect of prehospital emergency care, as timely and appropriate analgesia can substantially influence patient outcomes. Ideally, the selection and administration of analgesic agents should be guided by an objective assessment of pain severity. However, the present study’s absence of standardized pain assessment tools represents a significant methodological limitation. This limitation reflects a broader challenge in prehospital medicine, where numerous studies have reported persistently low rates of analgesia administration—27.5% overall and 40.7% specifically for injury-related pain [42,43]. Opioids are the most frequently used analgesics in prehospital settings (15.6%), followed by paracetamol (8.8%) and NSAIDs (2.1%) [42]. Specifically, in trauma cases, opioid administration is more prevalent (23.8%), while NSAID use significantly diminishes (0.2%) [43]. Similarly, Magnusson et al. (2021) reported that morphine was the most commonly administered analgesic, particularly for musculoskeletal injuries (80.7%), compared to those presenting with chest pain (35.4%) [30].

In this study, opioids were used extensively in patients with chest pain (Group 1), with the dosage directly correlating with diagnostic severity. Patients diagnosed with acute myocardial infarction (I21) received the highest doses of opioids, whereas those with non-specific chest pain (R07) were administered lower doses. In contrast, patients with suspected fractures (Group 2) were more likely to receive NSAIDs, even in cases of multi-region fractures where more potent analgesics would typically be indicated. Metamizole sodium was also frequently administered, and a combination of two analgesic agents was documented in only one case.

Despite these findings, several limitations of this study must be considered. A significant limitation of this retrospective study was the absence of documented pain intensity scores, even though pain assessment scales are commonly used. Data on the re-evaluation of pain intensity following therapy were also unavailable. While the brief average transport time in the study area (less than 15 min) might reduce the clinical need for repeated assessments, this data deficiency is noteworthy. Second, the absence of follow-up data on definitive hospital diagnoses and subsequent pain management protocols restricts a comprehensive understanding of the effectiveness of prehospital analgesia. Third, systemic barriers—including limited access to analgesic agents—further complicate effective pain management at the prehospital level. Although morphine is available, it is not universally accessible to all emergency medical providers. Furthermore, regulations restrict the use of more advanced analgesics, which might offer superior safety and efficacy profiles, especially in patients experiencing multi-trauma or more complex pain syndromes. These limitations suggest that current prehospital pain management protocols may be suboptimal, contributing to the continued prevalence of oligoanalgesia.

## 5. Conclusions

This study revealed that analgesic therapy was more frequently administered to patients presenting with chest pain, with opioid analgesics being the most commonly used agents. However, the frequency and dosage of analgesia varied significantly depending on the underlying diagnosis. Patients diagnosed with acute myocardial infarction (I21) most consistently received analgesics and at higher doses, whereas those with non-specific chest pain (R07) were less likely to receive analgesic therapy and typically received lower doses. In contrast, patients with suspected fractures were significantly less likely to receive analgesia, with NSAIDs being the most common choice. Among this group, pain management was most frequently provided to patients with suspected trunk fractures and least frequently to those with suspected cranial fractures. Notably, even patients with suspected fractures involving multiple anatomical regions received analgesia infrequently, and opioids were administered in fewer than 50% of these cases. These findings underscore persistent disparities in prehospital pain management and highlight the need for more consistent, diagnosis-informed analgesic protocols.

## Figures and Tables

**Table 1 biomedicines-13-01620-t001:** The demographic and clinical characteristics by group and in the total sample.

	Group 1	Group 2	Total	Significance
	N (%)			
Male ^$^	315 (62.6) ^a^	311 (45.3)	626 (100)	0.873
Female ^$^	188 (37.4)	375 (54.7) ^b^	563 (100)	***p* ˂ 0.001**
Age (years) ^&^	66.68 (15.7)	61.68 (22.04)	63.46 (19.18)	***p* ˂ 0.001**
Systolic blood pressure (mm Hg) ^#^	135 (120–160)	130 (120–140)	130 (120–150)	**0.013**
SpO_2_ ^#^	98 (96–99)	98 (96–98)	98 (96–99)	0.205
Heart rate ^#^	85 (71–100)	85 (75–98)	85 (73–99)	0.554
Total patients	503 (42.3)	686 (57.7)	1189 (100)	***p* ˂ 0.001**

^$^—chi-square test, ^a^—*p* < 0.001, ^b^—*p* < 0.01; ^&^—*t*-test of independence, mean (SD); ^#^—Mann–Whitney U test, median (P25–P75); N (%)—number (percentage); SD—standard deviation; P25–P75—25th percentile–75th percentile; bolded values are statistically significant.

**Table 2 biomedicines-13-01620-t002:** The distribution of analgesic therapy administration across groups and in the total sample.

	Group 1 ^$^ (N = 503)	Group 2 ^$^ (N = 686)	Total ^$^ (N = 1189)	Significance ^$^
Yes	303 (60.2) ^a^	123 (17.9)	426 (35.8)	***p* ˂ 0.001**
No	200 (39.8)	556 (81.1) ^a^	756 (63.6) ^a^	***p* ˂ 0.001**
Declined	0 (0)	7 (1.0)	7 (0.6)	

^$^—chi-square test; ^a^—*p* ˂ 0.000; bolded values are statistically significant.

**Table 3 biomedicines-13-01620-t003:** The frequency of analgesic administration by working diagnosis.

Group	Subgroups	Total	Analgesic Therapy		No Therapy		Declined	
		Number	Number	%	Number	%	Number	%
Group 1	Angina pectoris (I20) ^#^	220	120 ^ns^	54.5	100	45.5		
	Acute myocardial infarction (I21) ^#^	215	155 ^a^	72.1	60	27.9		
	Chest pain (R07) ^#^	68	28	41.2	40 ^ns^	58.8		
Group 2	Cranial fractures ^&^	14	1	7.1	13 ^a^	92.9		
	Trunk fractures ^#^	11	4	36.4	7 ^ns^	63.6		
	Extremity fractures ^#^	359	83	23.1	271 ^a^	75.5	5	1.4
	Multiple-region fractures ^#^	302	34	11.2	266 ^a^	88.1	2	0.7

^#^—chi-square test, ^&^—Fisher’s exact test: ^a^—*p* < 0.001; ^ns^—no statistical significance.

**Table 4 biomedicines-13-01620-t004:** The demographic and clinical parameters in Group 1 in relation to analgesic administration.

		Yes, N (%)	No, N (%)	*p*-Value
Group 1	Male ^$^	199 (63.2)	116 (36.8)	**<0.001**
	Female ^$^	104 (55.3)	84 (44.7)	0.145
	Age (years) ^&^	67.6 (13.8)	64.4 (17.9)	0.403
	Systolic blood pressure (mm Hg) ^#^	140 (120–160)	132.5 (118.8–150)	**0.021**
	SpO_2_ ^#^	98 (96–99)	98 (95.8–99)	0.997
	Heart rate ^#^	83 (71–100)	85 (71–100)	0.746
Subgroups				
Angina Pectoris	Male ^$^	83 (61.50)	52 (38.50)	**0.008**
	Female ^$^	37 (43.50)	48 (56.50)	0.233
	Age (years) ^&^	69.17 (14.06)	67.11 (18.36)	0.362
	Systolic blood pressure (mm Hg) ^#^	140 (120–160)	130 (110–145)	**0.021**
Acute Myocardial Infarction	Male ^$^	101 (71.10)	41 (28.90)	**<0.001**
	Female ^$^	54 (74.00)	19 (26.00)	**<0.001**
	Age (years) ^&^	65.95 (13.42)	67.92 (16.17)	0.626
Chest pain	Male ^$^	15 (39.50)	23 (60.50)	0.194
	Female ^$^	13 (43.30)	17 (56.70)	0.465
	Age (years) ^&^	65.14 (15.71)	61.53 (19.66)	0.421

^$^—chi-square test; N (%)—count (percentage); ^&^—independent *t*-test, mean (SD); ^#^—Mann–Whitney U test, median (P25–P75); SD—standard deviation; P25–P75—25th–75th percentile; *p*—significance; bolded values indicate statistical significance.

**Table 5 biomedicines-13-01620-t005:** The demographic and clinical parameters of Group 2 patients and subgroups in relation to analgesic administration.

		Yes, N (%)	No, N (%)	*p*-Value
Group 2	Male ^$^	51 (16.4)	260 (83.6)	**˂0.001**
	Female ^$^	72 (19.2)	303 (80.8)	**˂0.001**
	Age (years) ^&^	61 (17.6)	58.53 (16.2)	**0.026**
	Systolic blood pressure (mm Hg) ^#^	130 (118.8–150)	130 (120–140)	0.847
	SpO_2_ ^#^	98 (95.8–99)	98 (96–99)	0.916
	Heart rate ^#^	91.50 (80–102.5)	84 (73–97)	**0.049**
Subgroups				
Cranial fractures	Male ^$^	0	8 (100)	-
	Female ^$^	1 (16.70)	5 (83.30)	-
	Age (years) ^&^	NA	95.80 (3.03)	-
Trunk fractures	Male ^$^	2 (40.00)	3 (60)	-
	Female ^$^	2 (33.30)	4 (66.70)	-
	Age (years) ^&^	98.50 (0.71)	98.67 (0.58)	0.789
Extremity fractures	Male ^$^	28 (21.20)	104 (78.80)	**<0.001**
	Female ^$^	56 (24.70)	171 (75.30)	**<0.001**
	Age (years) ^&^	97.25 (1.98)	96.69 (3.09)	0.487
Multiple-region fractures	Male ^$^	21 (12.70)	145 (87.30)	**<0.001**
	Female ^$^	13 (9.60)	123 (90.40)	**<0.001**
	Age (years) ^&^	96.05 (4.47)	97.03 (3.21)	0.196
	Heart rate ^#^	96.50 (83–110)	88 (77–98)	**0.038**

^$^—chi-square test; ^&^—independent *t*-test; mean (SD); ^#^ Mann–Whitney U test; median (P25–P75); N (%)—number (percentage); SD—standard deviation; P25–P75—25th–75th percentile; NA—not available due to limited data; *p*—significance; bolded values indicate statistical significance; “-”—not computed due to data limitations or an insufficient sample size.

**Table 6 biomedicines-13-01620-t006:** The analgesic types and dosages administered in Group 1 patients.

Diagnosis	Drug	Number	Valid, %	%
Angina pectoris (I20) (120/220)	Tramadol, 150 mg	1	0.8	0.4
	Tramadol, 100 mg	50	41.7	22.7
	Tramadol, 50 mg	69	57.5	31.4
Acute myocardial infarction (I21) (155/215)	Morphine	1	0.6	0.4
	Tramadol, 100 mg	88	56.8	40.9
	Tramadol, 50 mg	66	42.6	30.7
Chest pain (R07) (28/68)	NSAID (ketoprofen)	1	3.6	1.5
	Tramadol, 100 mg	8	28.6	11.8
	Tramadol, 50 mg	18	64.3	26.5
	Tramadol, 50 mg + Metamizole Sodium	1	3.6	1.5

**Table 7 biomedicines-13-01620-t007:** The distribution of analgesic agents administered according to anatomical region among Group 2 patients.

Diagnosis	Drug	Number	Valid, %	%
Cranial fractures (1/14)	NSAID (ketoprofen)	1	100.0	7.1
Trunk fractures (4/11)	NSAID (ketoprofen)	2	50.0	18.2
	Tramadol, 100 mg	1	25.0	9.1
	Tramadol, 50 mg	1	25.0	9.1
Extremity fractures (83/359)	NSAID	32	38.5	8.9
	NSAID (ketoprofen)	21	25.3	5.8
	NSAID (diclofenac)	11	13.2	3.1
	Metamizole Sodium	19	22.9	5.3
	Paracetamol	1	1.2	0.3
	Tramadol, 100 mg	15	18.1	4.2
	Tramadol, 50 mg	16	19.3	4.4
Multiple-region fractures (34/302)	NSAID	15	44.1	5.0
	NSAID (ketoprofen)	8	23.5	2.6
	NSAID (diclofenac)	7	20.6	2.3
	NSAID (ketoprofen) + Tramadol, 50 mg	1	2.9	0.3
	Metamizole Sodium	3	8.8	1.0
	Tramadol, 100 mg	9	26.5	3.0
	Tramadol, 50 mg	6	17.6	2.0

## Data Availability

The raw data supporting the conclusions of this article will be made available by the authors upon request.

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
