# Peer review of "Variations in Prehospital Analgesic Use Based on Pain Etiology"

_biomedicines, 2025, doi:10.3390/biomedicines13071620_

Round 1
Reviewer 1 Report
Comments and Suggestions for Authors
Title
The title does not clearly reflect the nature of the study. How do the authors measure the impact? It is important to mention the study design and clearly state the research objectives.
Introduction
The introduction is engaging but does not clearly outline the knowledge gap that this study aims to address. Why is it important to understand the frequency of analgesic use in the prehospital setting?
Methods
Study population:
-
Group 1: Does this group include patients with a presumptive diagnosis of myocardial infarction or a confirmed diagnosis? Please clarify.
-
Group 2: Please include all corresponding meanings for the ICD-10 codes mentioned.
Statistical analysis:
Why did the authors choose to use Fisher’s exact test given the sample size? Please justify this decision.
Results
Why is the higher proportion of male patients in Group 1 relevant? Please explain its significance.
Table 2: p-values should not be reported as 0.000. Use "< 0.001" instead.
Discussion
It is recommended to begin the discussion by summarizing the main findings of the study in relation to the stated objectives.
Reviewer 2 Report
Comments and Suggestions for Authors
Dear Authors,
I have read the manuscript, and I send you my comments:
1) Methods: Page 3, line 112: Please explain why you chose diclofenac instead of ibuprofen or ketoprofen.
Moreover, why did you use ketorolac, which has no clinical indication for acute pain but only for postsurgical pain and for renal pain?
2) Please add ethics committee authorization.
3) Results: In Table 1 and Table 3, sex lines are missing.
4) In Results Table 5 and Table 6, please indicate the type of NSAID used.
5) Results Please provide each patient's characteristics for each treatment group.
6) Results: Please add the comorbidity of enrolled patients and also in patients for each group of age and of treatment.
7) Discussion: It may be revised in several sections; it seems to be a reproposition of the results.
Round 2
Reviewer 1 Report
Comments and Suggestions for Authors
The authors have made a great effort to improve the quality of their manuscript. Thank you very much
Author Response
Daar Reviewer,
Thank you very much
Reviewer 2 Report
Comments and Suggestions for Authors
Dear Authors,
I have read the manuscprit that has been imporved however I have 2 comments for you:
1) methods: please revise; it seems a prospective study with a division into 2 groups. It may be well clarified;
2) Results: Tab 4 and 5 must be revised these are very hard.
3) Discussion: please revise considering the results: what is the mechanism of pain in heart disease and in bose disease considering the difference in its perception in male and female?
Author Response
Dear Reviewer,
Please see the answers to your comments in the attached file and the changes made to the text in the attached manuscript.

Round 3
Reviewer 2 Report
Comments and Suggestions for Authors
none